# *Picea wilsonii* NAC31 and DREB2A Cooperatively Activate *ERD1* to Modulate Drought Resistance in Transgenic *Arabidopsis*

**DOI:** 10.3390/ijms25042037

**Published:** 2024-02-07

**Authors:** Yiming Huang, Bingshuai Du, Mingxin Yu, Yibo Cao, Kehao Liang, Lingyun Zhang

**Affiliations:** 1State Key Laboratory of Efficient Production of Forest Resources, College of Forestry, Beijing Forestry University, Beijing 100083, China; hyiming0928@163.com (Y.H.); dubingshuai624@163.com (B.D.); ymxbjfu@163.com (M.Y.); caoyibo@bjfu.edu.cn (Y.C.); lkhbjfu@163.com (K.L.); 2Key Laboratory of Forest Silviculture and Conservation of the Ministry of Education, College of Forestry, Beijing Forestry University, Beijing 100083, China

**Keywords:** *Picea wilsonii*, drought tolerance, *Arabidopsis*, NAC transcription factor, DREB2A

## Abstract

The NAC family of transcription factors (TFs) regulate plant development and abiotic stress. However, the specific function and response mechanism of NAC TFs that increase drought resistance in *Picea wilsonii* remain largely unknown. In this study, we functionally characterized a member of the PwNAC family known as *PwNAC31*. PwNAC31 is a nuclear-localized protein with transcriptional activation activity and contains an NAC domain that shows extensive homology with ANAC072 in *Arabidopsis*. The expression level of *PwNAC31* is significantly upregulated under drought and ABA treatments. The heterologous expression of *PwNAC31* in *atnac072 Arabidopsis* mutants enhances the seed vigor and germination rates and restores the hypersensitive phenotype of *atnac072* under drought stress, accompanied by the up-regulated expression of drought-responsive genes such as *DREB2A (DEHYDRATION-RESPONSIVE ELEMENT BINDING PROTEIN 2A)* and *ERD1 (EARLY RESPONSIVE TO DEHYDRATION STRESS 1)*. Yeast two-hybrid and bimolecular fluorescence complementation assays confirmed that PwNAC31 interacts with DREB2A and ABF3 (ABSCISIC ACID-RESPONSIVE ELEMENT-BINDING FACTOR 3). Yeast one-hybrid and dual-luciferase assays showed that PwNAC31, together with its interaction protein DREB2A, directly regulated the expression of *ERD1* by binding to the DRE element of the *ERD1* promoter. Collectively, our study provides evidence that PwNAC31 activates *ERD1* by interacting with DREB2A to enhance drought tolerance in transgenic *Arabidopsis*.

## 1. Introduction

Plants encounter a variety of challenging conditions throughout their growth and development, among which drought stress poses a significant challenge. Areas that are considered dry cover up to 6.1 billion hectares and occupy at least 41 percent of the Earth’s landmass [1]. Under drought stress, plants were found to undergo a range of changes, such as reduced hydraulic conductance in the stem and biomass in the shoot, resulting in severe water shortages and, ultimately, plant mortality [2]. The distribution of *Picea wilsonii*, a unique coniferous tree found in northern China, experiences significant shrinkage with the increased frequency of drought stress. However, our understanding of the response and regulatory mechanisms of plants under drought stress remains limited.

Plants have developed a complex set of self-regulatory mechanisms to mitigate adverse environmental impacts [3]. As one of these mechanisms, transcription factors (TFs) respond and initiate complex regulatory cascades to activate or inhibit the expression of target genes by binding to *cis*-elements of the transcription promoter regions. NAC (NAM, ATAF1/2 and CUC2) is a large family of TFs unique to plants responding to drought stress and consists of a C-terminal transcription activation domain that has a highly divergent and N-terminal NAC-binding domain that consists of nearly 150 amino acid residues [4,5]. Previous studies reported that the functions of a large number of NAC TFs in different plant species were involved in various processes, including growth [6] and development processes [7] and abiotic stress responses, such as salt and drought resistance [8,9,10]. For example, the overexpression of *RcNAC72* [11], *ThNAC4* [12], *FtNAC31* [10] and *AtruNAC36* [13] could enhance drought tolerance in *Arabidopsis*, and *GmNAC12* in *Glycine max* positively regulated plant tolerance to drought stress [14]. Plant survival rates were significantly improved in *ANAC019*-, *ANAC055*- and *ANAC072*-overexpressing *Arabidopsis* lines compared to the wild type [15]. In *Sorghum bicolor*, overexpressing *SbNAC9* lines showed that the drought tolerance of plants was improved accompanied by enhancements in the scavenging ability of reactive oxygen [16]. *GhNAC3* regulated the content of ABA to enhance the drought resistance of transgenic plants [17]. An *OsNAC092* mutant in rice protected cells from oxidant damage under drought stress [18]. Similar functions have been confirmed in woody plants. In *Malus pumila*, MdNAC1 enhanced drought tolerance in transgenic plants by promoting higher photosynthesis and activities of ROS-scavenging enzymes [19]. The overexpression of *PtNAC3* promoted reproductive success in plants under abiotic stress by shortening their lifespan [20]. In *Picea wilsonii*, *PwNAC2* and *PwNAC11* were also reported to play a positive role in the response to abiotic stress [3,21]. Nevertheless, not all NAC TFs function as positive regulators in plant abiotic tolerance responses. For example, ANAC069 inhibited the expression of downstream ROS()-scavenging genes by binding to the C[A/G]CG[T/G] sequence, ultimately resulting in an increase in sensitivity to osmotic stress [22]. Meanwhile, in our previous studies, we identified that PwNAC30 served as a negative regulator in response to drought and salt stress [23].

It is known that TFs can regulate the expression of target genes at the transcriptional and post-transcriptional levels by combining with the *cis*-acting elements present in the promoter regions. Previous NAC TFs studies showed that they can specifically bind to ABREs (ABA-responsive elements), DREs (dehydration-responsive elements), LTREs (low-temperature-responsive elements), MYB (myeloblastosis), MYC (myelocytomatosis), etc., to further regulate target genes that contain these *cis*-acting elements [24,25]. For example, in *sorghum*, *SbNAC9* directly activated the expression of the *SbC5YQ75* gene that encoded peroxidase, resulting in a reduction in drought tolerance [16]. OsNAC016 in *Oryza sativa* was found to interact with kinases GSK2 and SAPK8, thereby regulating plant architecture and drought tolerance [26]. In *Arabidopsis*, ANAC016 can repress *AREB1* transcription by specifically combining with the NAC016BM motif, ultimately reducing drought tolerance in plants overexpressing ANAC016 [27].

*Picea wilsonii*, a specific Chinese coniferous species, is known for its remarkable appearance and lush green crown, and has high ecological value [28]. Given its broad environmental adaptability, it can provide a good raw material for the timber industry [29]. However, with the increasing frequency of extreme weather, drought stress has emerged as a major environmental constraint on the growth and distribution of *P. wilsonii*. Therefore, it is necessary to uncover the molecular mechanisms underlying drought tolerance in *P. wilsonii*. In our previous reports, multiple *PwNAC* genes were identified under drought stress based on transcriptome data analysis [30], but little is known about their function. In this study, we identified a novel NAC TF, PwNAC31, from the RNA-seq database. PwNAC31 exhibits obvious upregulation in response to drought and ABA treatments. The heterologous expression of *PwNAC31* in *Arabidopsis* resulted in a substantial improvement in drought tolerance without affecting the growth and development of transgenic plants. In the present study, PwNAC31 could interact with DREB2A and bind to the DRE element to activate *ERD1* expression to improve drought tolerance in transgenic *Arabidopsis thaliana*. Our findings provide new insight into the molecular mechanisms of the NAC TF family governing drought responses in plants.

## 2. Results

### 2.1. Identification and Analysis of PwNAC31

Based on our previous RNA-seq analysis [30], the full-length coding region of PwNAC31 was amplified from cDNA of *Picea wilsonii*, which had an ORF (open reading frame) with 975 bp. Its corresponding protein contained 324 amino acids. The structural analysis showed that the molecular weight and PI of PwNAC31 were 36.83 kDa and 9.05, respectively. Furthermore, the protein sequences of PwNAC31 were grouped with other NAC TFs with identified functions by constructing a neighbor-joining phylogenetic tree (Figure 1A). Phylogenetic analysis showed that PwNAC31 from *Picea wilsonii*, exhibited a high evolutionary relationship with ANAC072 (At1g69490) from Arabidopsis. The analysis results of the protein multiple sequence alignment showed that PwNAC31 had a typical NAM domain (Figure 1B). The presence of the NAM domain in PwNAC31 proteins was highly homologous with the NAC protein sequences in other plant species.

### 2.2. PwNAC31 Responds to Drought Stress and ABA

It has been reported that the expression level of *NAC* genes is induced by drought and ABA treatments [31], so the relative quantitative assay of *PwNAC31* under drought and abiotic stress treatments was determined using RT-qPCR. Under drought treatment, *PwNAC31* was significantly induced in the late stages (3–6 h) (Figure 1C), and the expression level of *PwNAC31* was rapidly increased after 3–12 h of treatment with exogenous ABA (Figure 1D). Meanwhile, to investigate the tissue-specific expression pattern of *PwNAC31*, the total RNA was extracted from various *P. wilsonii*, and the results showed that *PwNAC31* was expressed in the pollen, root, stem, needle, seed and cone, but the fruits had the highest transcript level among all the tissues (Figure 1E). These results suggest that the expression of *PwNAC31* was dramatically induced in response to drought stress, and its regulation mechanism for drought tolerance might be related to ABA signals.

### 2.3. The Subcellular Localization of PwNAC31 and Activity of Transcriptional Activation

In order to exclude the effect of localization due to different GFP attachment positions, we constructed pCAMBIA120-PwNAC31, where the GFP fragment is located at the C-terminus of PwNAC31, and pSUPER1300-PwNAC31, where the GFP fragment is located at the N-terminal. Then, RACK1A-RFP and JAZ8-RFP were selected as the positive controls. The results showed that the fluorescence signals of PwNAC31-GFP and GFP-PwNAC31 were localized in the nucleus, which indicated that PwNAC31 was a nuclear protein (Figure 2A).

The yeast two-hybrid assay was conducted to detect whether PwNAC31 had transcriptional activating activity. The full-length, C-terminal and N-terminal regions of PwNAC31 were combined with pGBKT7 plasmids, respectively, and then, transformed into yeast AH109 cells. AH109 cells containing a pGBKT7 empty vector were chosen as negative controls, and those containing a pGBKT7-ANAC092 recombinant plasmid were chosen as the positive controls. All of the transformants could grow normally on the Trp-deficient medium. Furthermore, different concentrations of positive clones were dotted on the SD/Trp-His-Ade-selective medium. The results showed that yeast strains containing pGBKT7-PwNAC31-C, pGBKT7-PwNAC31 and pGBKT7-ANAC092 grew well and turned blue on X-gal filters, yet the AH109 cells containing a pGBKT7 empty vector and pGBKT7-PwNAC31-N could not grow on the same plates, indicating that both full-length and C-terminal of PwNAC31 had transcription-activating activity (Figure 2B). In addition, the yeast two-hybrid assay showed that PwNAC31 could not form a homodimer by itself (Figure 2C).

### 2.4. PwNAC31 Enhances Seed Vigor and Germination under Drought Stress

In order to explore the function of PwNAC31 in plants, *35S::PwNAC31* in the *Arabidopsis atnac072* mutant was heterogeneously expressed, and two homozygous T3 lines (RE2, RE4) with high *PwNAC31* expression levels were obtained using RT-qPCR detection (Appendix A). The seeds of WT, *atnac072*, RE-2 and RE-4 were sown on MS medium containing different concentrations of mannitol to observe and compare their germination. The RE lines did not show a significant difference in WT and mutant seeds under the control conditions. Under the medium containing 100 mM mannitol, the germination rates of the WT and the RE lines were close to 60% on the 4th day, whereas for the mutant, it was less than 40% (Figure 3A,B). However, after a 7-day period of drought stress, the final germination percentages of the WT and RE lines were more than 80%, while that of the mutant line was around 60%. Similarly, with an increase in mannitol concentration (200 mM), the germination percentages of the WT and RE lines after 7 days were about 60%, while the mutant line was only 40% (Figure 3C). These results suggest that PwNAC31 conferred upon the seeds more vigor and higher germination rates under drought stress.

### 2.5. PwNAC31 Enhances Drought Tolerance in Transgenic Arabidopsis

To further confirm the function of PwNAC31 and its effect on plant growth, we grew *Arabidopsis* WT, an *atnac072* mutant and two transgenic RE lines of PwNAC31 on MS medium supplemented with 100, 200 and 300 mM mannitol (Figure 4A,C). Under 100 mM mannitol culture conditions, the mean values of root length in the WT line increased by 34%, and the averages of root lengths in the RE lines increased by at least 38% compared to *atnac072*. On the MS medium with 200 mM mannitol, the root lengths of RE-2, RE-4 and WT were 22%, 21% and 21% longer, respectively, than that of *atnac072*. Meanwhile, there was no significant difference in the length of the WT and RE lines under 100 and 200 mM mannitol. Under 300 mM mannitol culture conditions, the seeding growth of WT, RE and *atnac072* was inhibited due to the excessive concentration of mannitol. These results indicate that PwNAC31 can restore the mutant phenotype under mannitol treatments.

Similarly, for adult transgenic *A. thaliana*, all lines showed a consistent phenotype under normal growth conditions. Most leaves of the mutant lines suffered severe wilting and even died, and were unable to recover after re-watering (Figure 4B). In contrast, the WT and RE lines showed slight wilting and recovered well after 3 days of re-watering, with a final survival rate of 80% compared to the mutant line, which achieved less than 20% (Figure 4D). These results suggest that PwNAC31, as a positive regulator, could significantly increase tolerance to drought stress in transgenic *Arabidopsis*. After three hours of drought treatment, the relative leaf water content of the mutant lines was significantly reduced by 40% compared to the WT and RE-2 and 4 lines. The drought treatment resulted in a general decrease in chlorophyll content compared to the control, with the mutant having the lowest content (Figure 4E,F).

### 2.6. PwNAC31 Interacts with ABF3 and DREB2A

In order to reveal the possible regulatory mechanisms of *PwNAC31* responding to drought stress, the potential interaction proteins of PwNAC31 were predicted by the STRING website (https://cn.string-db.org/) accessed on 20 May 2022, among which DREB2A, ABF3, RHA2A, RHA2B and ATHB-7 were selected to confirm the hypothesis. The combined plasmids of AD-PwNAC31 and BD-ABF3, and AD-PwNAC31 and BD-DREB2A, showed the normal growth of yeast cells on four amino acid-deficient media, suggesting that PwNAC31 was able to interact with DREB2A and ABF3 in yeast (Figure 5A). This finding was further demonstrated with a BiFC assay in *N. benthamiana* leaves by co-expressing PwNAC31 and ABF3 or DREB2A. The fluorescence signals could be observed in the nuclei, indicating that PwNAC31 interacted with both ABF3 and DREB2A in vivo (Figure 5B).

### 2.7. Expression Pattern Analysis of Genes Responding to Drought and ABA in PwNAC31 Transgenic Arabidopsis

Based on the regulatory network of homologous genes of PwNAC31, the expression level of genes responding to drought and ABA pathway-related stress, including *ERD1*, *DREB2A*, *RD29A*, *ABI5*, *ABF3* and *NCED3*, were examined using RT-qPCR in different *Arabidopsis* lines after PEG treatment (Figure 6). The results showed that the relative expression of these genes remained low across all lines under normal conditions. The results showed that the expression levels of the drought-responsive genes and ABA-signaling-pathway genes were up-regulated to various extents in the WT and RE lines after PEG treatment compared to the *atnac072* mutant. Especially, the expression levels of *DREB2A* peaked and were twice as high as that of the mutant line. After 6 h of PEG treatment, the expression level of *ERD1* trebled in the mutant line. These results suggested that *PwNAC31* might enhance drought tolerance by regulating the expression of stress-related genes.

### 2.8. PwNAC31 Can Combine with the Promoter Region of DREB2A and ERD1

To further reveal the relationship between PwNAC31, *ERD1* and *DREB2A*, we performed Y1H assays. Compared to the negative control, yeast cells containing combined plasmids of AD-PwNAC31 and pAbAi-*ERD1* pro, and AD-PwNAC31 and pAbAi-*DREB2A* pro, could grow normally on the SD/Ura-Leu + AbA-selective medium (Figure 7B), indicating that PwNAC31 can combine with the *ERD1* and *DREB2A* promoters. Further analysis of the *ERD1* and *DREB2A* promoter sequences revealed that they contained a number of *cis*-acting elements involved in the plant stress response, including the DRE, ABRE and MYC elements (Figure 7A). To test the potential binding sites on the *ERD1* and *DREB2A* promoter with PwNAC31, short tandem repeats of DRE, ABRE and MYC elements were, respectively, constructed into the pAbAi vector, and transformed into yeast cells together with the recombinant plasmid AD-PwNAC31. Compared to the negative control, only yeast cells containing combined plasmids of AD-PwNAC31 and pAbAi-MYC or -DRE elements could normally grow on the selective medium, demonstrating that PwNAC31 specifically binds to the DRE and MYC elements, but not the ABREs (Figure 7C).

In addition, it has been reported that DREs in the *ERD1* promoter are binding sites of DREB2A [21]. Therefore, we performed a dual-LUC assay to determine whether PwNAC31 can synergistically enhance the transcriptional activation of the *ERD1* gene when co-expressed with DREB2A. As shown in Figure 7D, the expression of PwNAC31 alone could increase the activation of the reporter gene in tobacco. In addition, the simultaneous expression of PwNAC31 and DREB2A further increased promoter activity. Overall, these results indicated that PwNAC31 controlled the expression of ERD1 as a positive regulator of *ERD1* expression, and its co-expression with DREB2A enhanced this activation by binding to the DRE motifs. For ABF3, as another interacting protein of PwNAC31, we further detected whether it can regulate the expression of *ERD1* together with PwNAC31 using a dual-LUC assay. The results showed that the *ERD1* promoter was not activated when ABF3 acted independently or when PwNAC31 and DREB2A were present together (Appendix A).

## 3. Discussion

NAC proteins, as transcription factors, are known to contain a highly conserved NAC domain and play an important role in flower formation and fruit ripening [32], roots growth [33], leaf senescence [34], low/high-temperature stress [35,36] and drought and flooding stress [37,38,39]. In the present study, we investigated the function and mechanism of *PwNAC31* in the response to drought stress in plants. *PwNAC31* was identified as a homologous gene of the *Arabidopsis* NAC transcription factor ANAC072 through protein multiple sequence alignment and BLAST analysis. Thus, the *Arabidopsis* mutant *atnac072* was chosen for the transformation experiment to explore the possible role of PwNAC31. Here, we found that PwNAC31 was a nuclear-localized protein with transcriptional activation activity and contained a typical NAC domain. The transcript level of *PwNAC31* showed a significant increase in *P. wilsonii* seedlings under drought stress and ABA treatments. Previous studies have reported that the expression of *ANAC072* was induced under drought stress in *Arabidopsis* [15]. Consistent with these findings, the transcript level of the homologous gene *PwNAC31* displayed a great increase when *P. wilsonii* seedlings were exposed to drought stress, indicating its potential role in response to abiotic stress. Furthermore, we found that the heterologous expression of *PwNAC31* can largely restore the drought sensitivity of the *atnac072* mutant both for seedlings and adult plants. Under drought treatment, the WT and *PwNAC31*-RE lines conferred upon the seeds more vigor and higher germination rates compared with the mutant (Figure 3 and Figure 4A). Meanwhile, the *PwNAC31*-RE lines showed higher levels of relative leaf water content and chlorophyll content compared with the mutant (Figure 4E,F). In this study, we observed no significant differences in the growth of seedlings, the flowering phase and the size of the rosette leaves between the PwNAC31 RE lines, the WT and the *anac072* lines (Appendix A). These results demonstrate that PwNAC31 functions as a positive regulator in response to drought stress in plants.

NAC TFs can bind to the promoter of target genes to improve plant tolerance to abiotic stress [24]. Therefore, it is important to explore the potential regulatory mechanism of NAC TFs by identifying their target genes. In *Musa paradisiaca*, *MusaNAC042* could activate the expression of *CBF/DREB* to improve the capability of drought [40]. A recent report found that the SbNAC9 could enhance the scavenging ability of reactive oxygen and activate the expression of the stress-responsive genes *SbDREB1A, SbDREB2A* and *SbNCED3* to improve the drought tolerance of *Sorghum bicolor* [16]. In our study, PwNAC31 could interact with the DREB2A and ABF3 in the Y2H assay, and the BiFC assay demonstrated the interaction between PwNAC31 and ABF3 or DREB2A in vivo (Figure 5), and the expression level of DREB2A was significantly increased in *PwNAC31* transgenic *Arabidopsis* under drought stress. Based on these results, we speculated that drought resistance in PwNAC31 transgenic *Arabidopsis* was enhanced through the interaction between PwNAC31 and DREB2A. It was reported that the DREB family of transcription factors could rapidly respond to external signals to participate in the control of osmotic stress [41], and the promoter region of DREB2A contained the ABRE and MYC elements, which could bind to NAC TFs [15,21,42,43]. The same results were also obtained in our study. PwNAC31 could bind to the MYC motif on the *DREB2A* promoter to form a complicated network, participating in the control of their target genes’ expression.

Notably, a significant difference in the expression of *ERD1* was also found in the mutant and the RE lines of transgenic *Arabidopsis* (Figure 6). ANAC019, ANAC055 and ANAC072 are three drought-induced NAC-like transcription factors that bind to the promoter region of the *ERD1* gene and positively regulate the plant response to drought [15]. The co-over-expression of the *ZFHD1* and *NAC* genes induced the expression of the *ERD1* gene in *Arabidopsis* transgenic plants [44]. In a previous study, ANAC072 was engaged in regulating plant responses to early drought signals by binding to the DRE element on the promoter of the *ERD1* gene [15]. Our results found that PwNAC31 also combined with the promoter of *ERD1*. Furthermore, we constructed tandem repeats containing the ABRE and DRE elements and found that PwNAC31 could specificity bind to DREs but not ABREs, indicating that PwNAC31 was likely to be involved in the regulation of drought-responsive genes. This differs from another NAC member, PwNAC11 [21], which could combine with ABREs instead of DREs to activate the expression of the downstream gene *ERD1* under abiotic stress, implying that this is an ABA-dependent signaling cascade controlled by PwNAC11. A similar regulatory mechanism of the NAC family was also found in other plant species, such as in *Glycine max*; GmNAC20 regulated stress tolerance through the DREB/CBF–COR pathway, where GmNAC11 more likely regulated DREB1A and other genes responding to stress, and furthermore, responded to abiotic stress [45].

In addition, PwNAC31 was significantly induced by ABA signals, and our results also confirmed the interaction between PwNAC31 and ABF3. The ABF transcription factors are crucial for ABA signaling, and function downstream of various ABA-mediated stress responses [46,47]. Nevertheless, the expression of PwNAC31 more strongly promoted the activation of the reporter gene than the empty vector in tobacco, while ABF3 poorly activated LUC expression. Furthermore, the simultaneous expression of PwNAC31 and ABF3 did not enhance the activation of the ERD1 promoter more than PwNAC31 alone. These results suggest ABF3 plays an extremely limited role in the molecular mechanisms of PwNAC31’s response to drought by activating the expression of *ERD1*. Combined with the results that there was significant upregulation of the ABA-signaling-pathway genes *ABI5* and *ABF3* in *PwNAC31* transgenic *Arabidopsis* plants after exposure to drought stress, we speculate that NAC31 may also be regulating the drought tolerance mechanism through the ABA-dependent pathway, with ABF3 or ABI5 involved in this mechanism.

Based on these results, we sketched the regulatory mechanism of the drought response pathway mediated by PwNAC31 (Figure 8). First, PwNAC31 responds to drought and can, respectively, bind to the MYC and DRE motifs on the promoters of *DREB2A* and *ERD1* to form a complicated network. Then, PwNAC31 and DREB2A can activate the expression of *ERD1* at the transcriptional level. The protein interaction between PwNAC31 and DREB2A strengthens the transcription of *ERD1* under drought stress. In short, the findings of these synergistic pathways provide a theoretical basis and candidate genes for improving drought tolerance in *Picea wilsonii*.

## 4. Materials and Methods

### 4.1. Plant Materials and Drought Treatments

The *P. wilsonii* seeds (from Dunhua Forestry Bureau, Jilin, China) were sown on moistened filter paper in glass Petri dishes and cultured in a light incubator at 21 °C during 16 h of light and 8 h of darkness. Then, about 8 weeks after germination, the seedlings were transferred to nutrient soil for further growth. For the drought treatment, the seedlings of *P. wilsonii* were placed on absorbent paper without water to simulate drought stress; then, the whole plants were harvested at 0, 3, 6 and 12 h, respectively. For ABA treatment, the seedlings were treated with 100 µM ABA solution (Coolaber Science & Technology Co., Ltd., Beijing, China), and then, collected and frozen for 0, 3, 6 and 12 h, respectively. Different tissues, including pollen, root, stem, needle and cone, were collected from the mature *P. wilsonii* trees in the Institute of Botany, the Chinese Academy of Sciences. The primers used in this study are shown in Appendix A.

*Arabidopsis thaliana* Columbia-0 wild-type (WT) and *atnac072* mutant (obtained from the *Arabidopsis* Biological Resource Center, line salk_063576) were used in this study to investigate the function of *PwNAC31*. *Agrobacterium tumefaciens* GV3101 carrying pCAMBIA1205-PwNAC31 was used for *Arabidopsis* transformations. The T-DNA of the binary vector, pCAMBIA1205, contains the GFP reporter gene under the control of the cauliflower mosaic virus (CaMV) 35S promoter and the *CAT* gene that confers resistance to the antibiotic chloramphenicol. The T1 seeds were collected and spread on MS medium agar plates containing 50 mg/mL chloramphenicol (Sigma-Aldrich Trading Co., Ltd., Shanghai, China) and 100 mg/mL hygromycin (Sigma-Aldrich Trading Co., Ltd., Shanghai, China). After 14 days of selection in medium, seedlings were transplanted to nutrient soil with vermiculite and grown to maturity in a greenhouse. Transformants were allowed to self-fertilize, and T2 seeds were collected individually from each Tl plant. Then, T3 homozygous generations were obtained whose seeds all grew normally on the selection medium. The seeds from the T3 homozygous generations of the restoration lines were collected for further experiments. Then, the sterilization, sowing and culture of the WT, *atnac072* mutant and PwNAC31 restoration lines seeds were performed as described previously [21]. On the 14th day, the seedlings were moved to the soil for further growth.

Furthermore, we used different concentrations of mannitol (Sigma-Aldrich Trading Co., Ltd., Shanghai, China) (0, 100, 200 and 300 mM) to simulate drought stress conditions for the research concerning the influence of drought on germination rates in *Arabidopsis* seeds. The germination rate of the seeds needed to be counted every day, considering exposure of the germs to air as the criterion for germination. For the root length experiment, one-week-old seedlings were spread on the same medium as above; after 10 days of growth, the phenotype of the seedlings was observed and photographed. To assess the drought tolerance in the case of a lack of water, all of the seedlings were transferred the same amount of soil mixture 2 weeks after planting, and they were subjected to drought stress without water for 15 days. Then, after re-watering for 3 days, the survival rate of the seedlings was recorded.

### 4.2. Gene Isolation and Homologous Analysis

The full-length coding region of *PwNAC31* was amplified from cDNA in *Picea wilsonii* seedlings, which was then blasted in the NCBI database (National Center for Biotechnology Information, https://www.ncbi.nlm.nih.gov/) accessed on 6 June 2020 to obtain its homologous proteins. All of the sequences that we screened were analyzed with ClustalX and used to construct a phylogenetic tree via the neighbor-joining method using MEGA5 (11.0.13) software [48]. The primers used in this study are shown in Appendix A.

### 4.3. RNA Isolation and Quantitative Real-Time PCR

Total RNA was extracted from *P. wilsonii* and *Arabidopsis* under various treatments according to the manual of a plant RNA extraction kit that was purchased from Kangwei Century Biotechnology Co., Ltd., Beijing, China. The cDNA was synthesized using the total RNA (1 µg) after being measured using a Nanodrop 2000 spectrophotometer (Thermo Fisher, Waltham, MA, USA) as the template, and the reaction process was performed according to the manual of the AccuRT Genomic DNA removal kit (Applied Biological Materials Inc., Vancouver, BC, Canada).

To analyze the expression of the target genes, SYBR Green Master Mix enzyme (by ABI, Vernon, CA, USA) was used for the RT-qPCR reactions. *PwEF-1α* and *Atactin* were chosen as reference genes in *P. wilsonii* and *Arabidopsis*, respectively, which could be stably expressed under the drought stress and ABA treatments. Finally, the relative expression levels of the target genes were calculated using the 2^−∆∆CT^ method. The primers used in this study are shown in Appendix A.

### 4.4. Subcellular Localization

To exclude the possible effect of GFP on localization, PwNAC31-ORF was constructed in the pCAMBIA1205 vector, in which GFP was located at the N-terminal end of the target fragment (designated as GFP-PwNAC31), and the pSuper1300 vector, in which GFP was located at the C-terminal end of the target fragment (designated as PwNAC31-GFP). GFP-PwNAC31 and PwNAC31-GFP were individually injected into the *Nicotiana benthamiana* epidermal cells; the empty 35S::GFP, pcDNA3.1-RACK1A-RFP and pcDNA3.1-JAZ8-RFP, as with the controls, were also co-transferred into the leaves with the corresponding recombined vector containing PwNAC31. The T-DNA of the binary vector, pcDNA3.1, contains the RFP reporter gene under the control of the CaMV35S promoter. RACK1A was reported to be localized in the cytoplasm, cell membrane and nucleus [49], and JAZ8 was reported to be localized in the nucleus [50]. After 48 h, the GFP fluorescence was observed under a confocal laser microscope (Leica TCS SP5, Germany) with excitation and emission wavelengths of 488 nm and 515 nm, respectively, and the RFP fluorescence was observed with excitation and emission wavelengths of 555 nm and 583 nm, respectively.

### 4.5. Transcriptional Activity Analysis and the Verification of Interaction Proteins via Y2H Assay

The transcriptional activity of PwNAC31 was detected by means of the pGBKT7 [4] vector. Firstly, the coding region of PwNAC31 was cloned into the BD vector. Then, PwNAC31-BD was transformed into yeast AH109 cells (Shanghai Weidi Biotechnology Co., Ltd., Shanghai, China). Different concentrations of positive clones that could normally grow in the SD/-Trp medium were dotted on the SD/-Trp/-His-Ade + X-α-gal medium (Coolaber Science & Technology Co., Ltd., Beijing, China). After incubation at 30 °C for 3 d in the incubator, the growth situation of the yeast cells was photographed and recorded.

PwNAC31 and its predictive interaction proteins were verified using the Y2H (yeast two-hybrid) assay, which was performed as previously described [51]. Briefly, the PwNAC31 CDS was cloned into the pGADT7 (AD) vector as bait, and the coding regions of *PwNAC31*, *DREB2A*, *RHA2A*, *ATHB* and *RHA2B* were constructed in the pGBKT7 [4] vector. The PwNAC31-AD with recombinant BD vectors were individually co-transformed into yeast AH109 cells and selected on SD/-Trp-Ura-His-Ade + X-α-gal medium. It was noted that the co-transfection of PwNAC31-AD with PwNAC31-BD was used to detect if PwNAC31 formed a homodimer by itself. Primers used in this study are shown in Appendix A.

### 4.6. The Verification of Interaction Proteins by BiFC (Bimolecular Fluorescence Complementation)

The PwNAC31-ORF was constructed in the pSPYCE (cYFP) vector when creating the fusion protein PwNAC31-cYFP. The full lengths of *DREB2A* and *ABF3 CDS* were cloned into the pSPYNE (nYFP) vector, respectively. pSPYCE- and pSPYNE-expressing strains were used as negative controls. Then, the recombinant vectors were transiently transformed into transgenic tobaccos which contained a red fluorescent protein emitting a nuclear localization signal (NLS-mCherry) [51]. After 3 days, the fluorescence signal was observed under a confocal laser microscope (Leica TCS SP5, Germany) with excitation and emission wavelengths of 488 nm and 515 nm, respectively. The primers used in this study are shown in Appendix A.

### 4.7. Yeast One-Hybrid (Y1H) Assays

The full-length coding sequence of *PwNAC31* was combined into the AD vector, and the promoter sequences of *DREB2A* and *ERD1* and short tandem repeats of the ABRE, DRE and MYC motifs were individually cloned into the pAbAi vector. The PwNAC31-AD and recombinant pAbAi vectors were simultaneously inserted into yeast AH109 cells, and then, we selected SD/-Ura-Leu + AbA as the medium. pGADT7-P53 and pAbAi-P53 were used as the positive controls.

### 4.8. Dual-Luciferase Assay

The *ERD1* and *DREB2A* promoters were, respectively, linked to a reporter vector containing the *REN* gene and *LUC* gene pGreenII 0800-LUC vector as a reporter vector under the control of the CaMV35S promoter. The CDS of *PwNAC31* and DREB2A were constructed in a pGreenII 62-SK vector driven by a 35S promoter as an effective carrier, and renilla luciferase (rLUC) activity was used as a reference. The firefly luciferase gene (fLUC) was initiated and expressed by the *ERD1* and *DREB2A* promoters. The instantaneous expression of combined recombinant plasmid in tobacco was performed as the above. The effect of PwNAC31 on the *ERD1* and *DREB2A* promoters was determined by detecting the chemiluminescence values of fLUC and rLUC using the Dual Luciferase Reporter Gene Assay Kit (by Beyotime, Shanghai, China) via a luminometer (Lumat LB 9501 by Berthold, Bad Wildbad, Germany). The ratio of the fLUC chemiluminescence value to the rLUC chemiluminescence value was LUC/REN. The primers used in this study are shown in Appendix A.

### 4.9. Physiological Index Measurement

Fruit pod length and plant height of *Arabidopsis* seedlings were measured after four weeks of growth in soil. Several stress damage indices were assessed on seedlings in order to evaluate the effects of drought treatment. Drought-stressed seedlings were weighted at periods of 0, 0.5, 1, 2, and 3 h at room temperature in order to calculate the relative leaf water content. We used 80% acetone to extract the chlorophyll content, which was determined with reference to the method of Shah et al. [52].

### 4.10. Statistical Analysis

The data obtained from three replicates were presented as means ± SD. Variance analysis was performed using SPSS software version 20.0. Duncan’s multiple range test was performed for ANOVA analysis, with a significance level of *p* < 0.05 considered to be statistically significant compared to the control. Graphs were generated using Sigma Plot 10.0. All of the above experiments had three or more biological replicates.

## Figures and Tables

**Figure 1 ijms-25-02037-f001:**
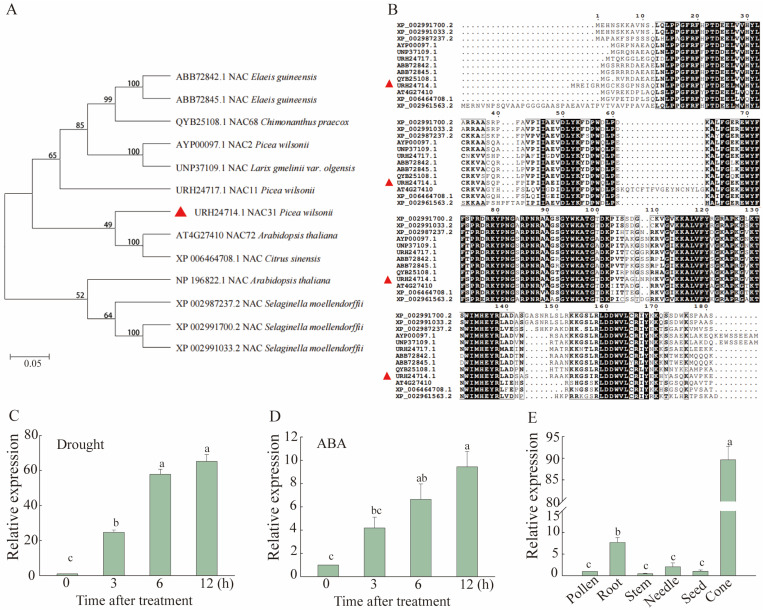
Multiple sequence alignment and analysis of PwNAC31. (**A**) Comparison of genetic relationships between PwNAC31 (marked with a red triangle) and 12 other NACs. (**B**) Homology analysis of protein sequences between PwNAC31 and 12 other NACs. (**C**,**D**) Expression profiles of *PwNAC31* in *P. wilsonii* seedlings under drought stress and ABA treatments. (**E**) Expression levels of *PwNAC31* in different tissues. The error bars indicate ±SD of three independent replications, and the different lowercase letters in Figure (**C**–**E**) mean significant differences at 5% level between treatments.

**Figure 2 ijms-25-02037-f002:**
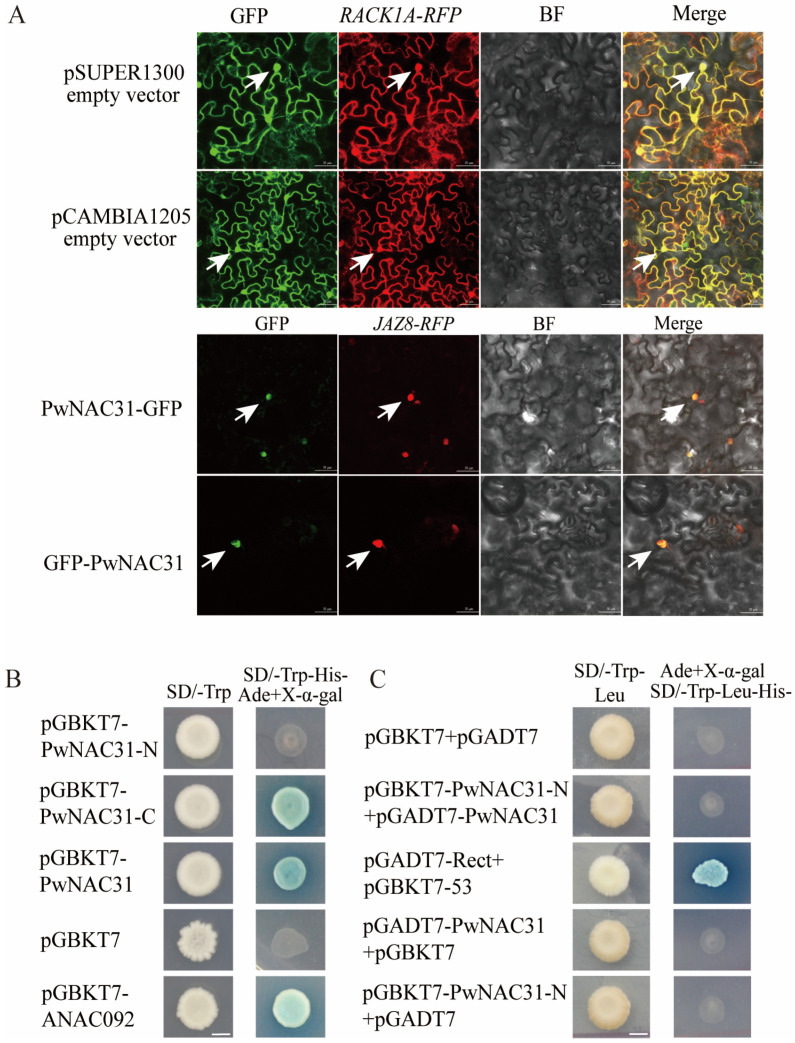
Subcellular localization and transactivation analysis of PwNAC31. (**A**) Subcellular localization of PwNAC31. RACK1A is a protein localized in the nucleus, cytoplasm and cell membrane, while the marker protein JAZ8 is localized in the nucleus. PwNAC31-GFP expresses the GFP tag at the C-terminus, and GFP-PwNAC31 expresses the GFP tag at the N-terminus. The white arrows point to nuclei. The bar in each picture represents 50 μm. (**B**) Yeast transactivation activity assay. pGBKT7-ANAC092 was used as a positive control. pGBKT7-PwNAC31-N refers to the PwNAC31 with the N-terminal and pGBKT7-PwNAC31-C refers to the PwNAC31 with the C-terminal. (**C**) The yeast two-hybrid assay was performed to determine whether PwNAC31 can form homodimers. pGBKT7 + pGBDT7 was used as a negative control, and pGBDT7-Rect + pGBKT7-53 as a positive control. The bar in each picture represents 5 mm.

**Figure 3 ijms-25-02037-f003:**
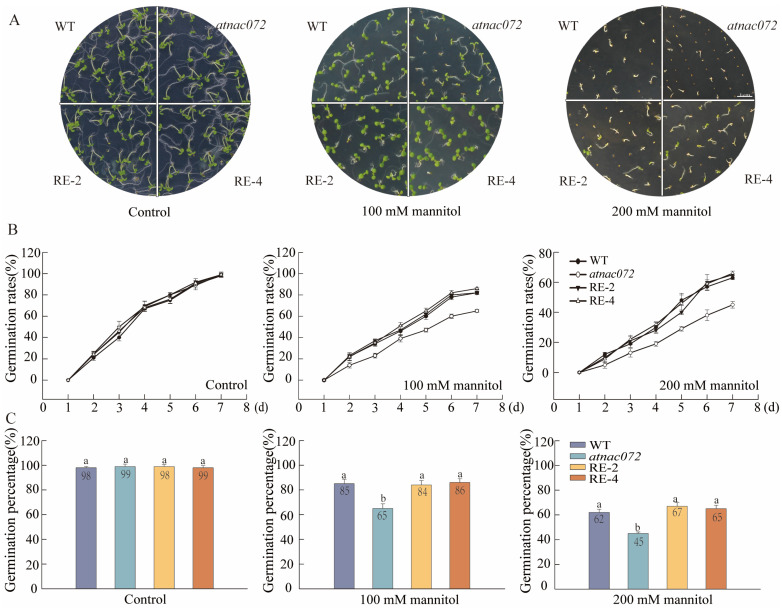
The impact of PwNAC31 on seed germination in *Arabidopsis* under simulated drought stress conditions. (**A**) Germination observation of WT, *atnac072*, RE-2 and RE-4 T3 homozygous transgenic seeds under control conditions (without treatment) and 100 mM and 200 mM mannitol after 7 days, with scale bar of 5 mm. (**B**) Germination rates and (**C**) germination percentages of different lines under control conditions (without treatment) and 100 mM and 200 mM mannitol for 7 days were determined. Error bars indicate ±SD of three independent replications, and the different lowercase letters in Figure (**C**) mean significant differences at 5% level between treatments.

**Figure 4 ijms-25-02037-f004:**
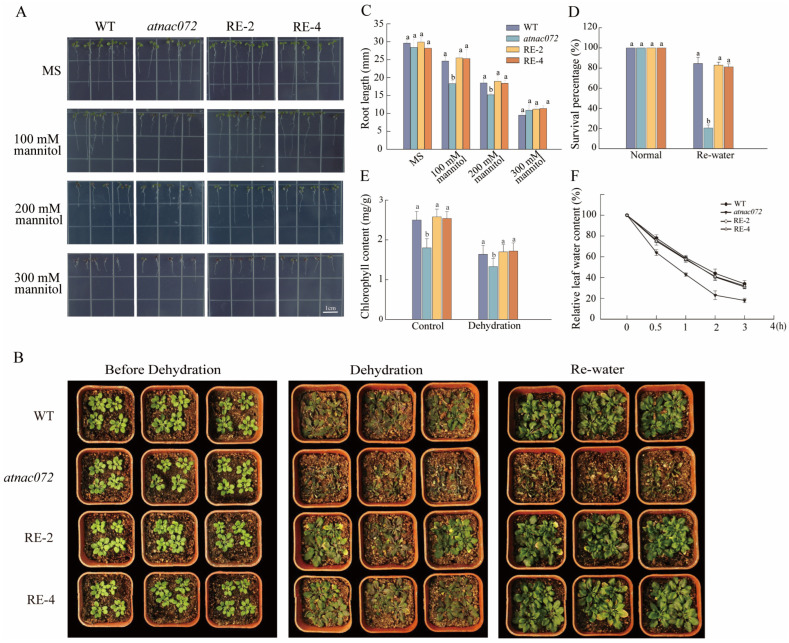
PwNAC31 enhances drought tolerance in transgenic *Arabidopsis*. (**A**) The root elongation phenotype of WT, *atnac072*, RE-2 and RE-4 T3 homozygous transgenic lines under control condition and 100 mM, 200 mM and 300 mM mannitol treatments. The bar in each picture represents 1 cm. (**B**) Phenotype of all seedlings of *Arabidopsis* lines cultured in soil before drought stress, after drought stress and re-watering. (**C**) Root length statistics for different *Arabidopsis* lines treated with mannitol. Scale bar = 1 cm. (**D**) Quantifying the survival percentage under normal conditions and re-watering treatment. (**E**) Chlorophyll contents and (**F**) relative leaf water content of different *Arabidopsis* lines after dehydration. The error bars indicate ±SD of three independent replications, and the different lowercase letters in Figure (**C–E**) mean significant differences at 5% level between treatments.

**Figure 5 ijms-25-02037-f005:**
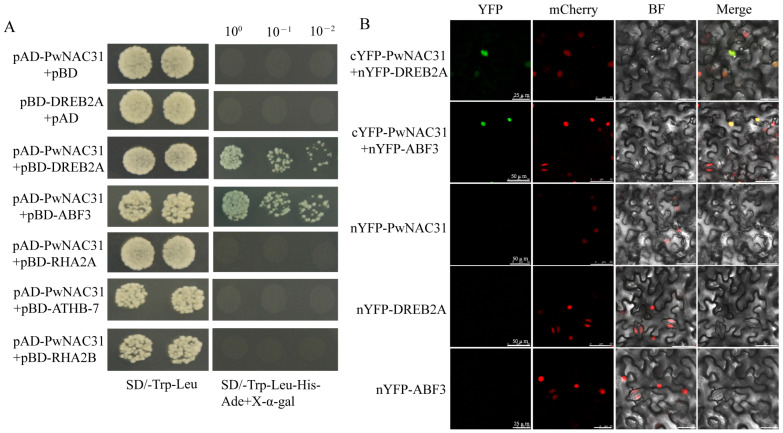
PwNAC31 interacts with DREB2A and ABF3. (**A**) Yeast two-hybrid assays for PwNAC31 with DREB2A, ABF3, RHA2A, ATHB-7 and RHA2B. The yeast strains containing AD-PwNAC31+ BD and AD + BD-DREB2A fusion plasmids were used as the negative control. (**B**) BiFC assays for the interactions between PwNAC31 and DREB2A/ABF3 in transgenic tobacco plants containing red fluorescent protein that can emit a nuclear localization signal (NLS-mCherry). The white bars represent 50 μm and 25 μm.

**Figure 6 ijms-25-02037-f006:**
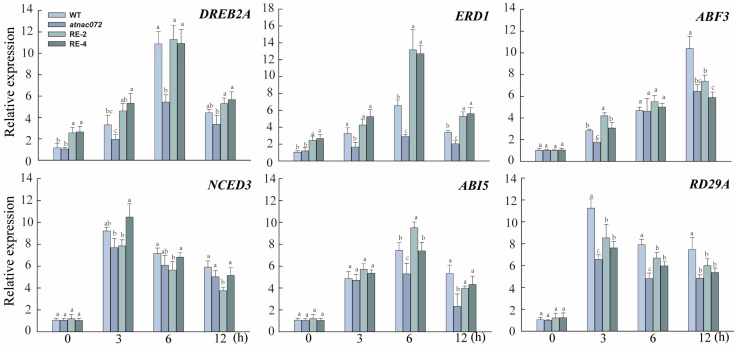
Expression levels of key stress-responsive genes and ABA-responsive genes in WT, *atnac072* mutant and PwNAC31-RE lines after PEG treatment. The error bars indicate ±SD of three independent replications, and the different lowercase letters mean significant differences at 5% level between treatments.

**Figure 7 ijms-25-02037-f007:**
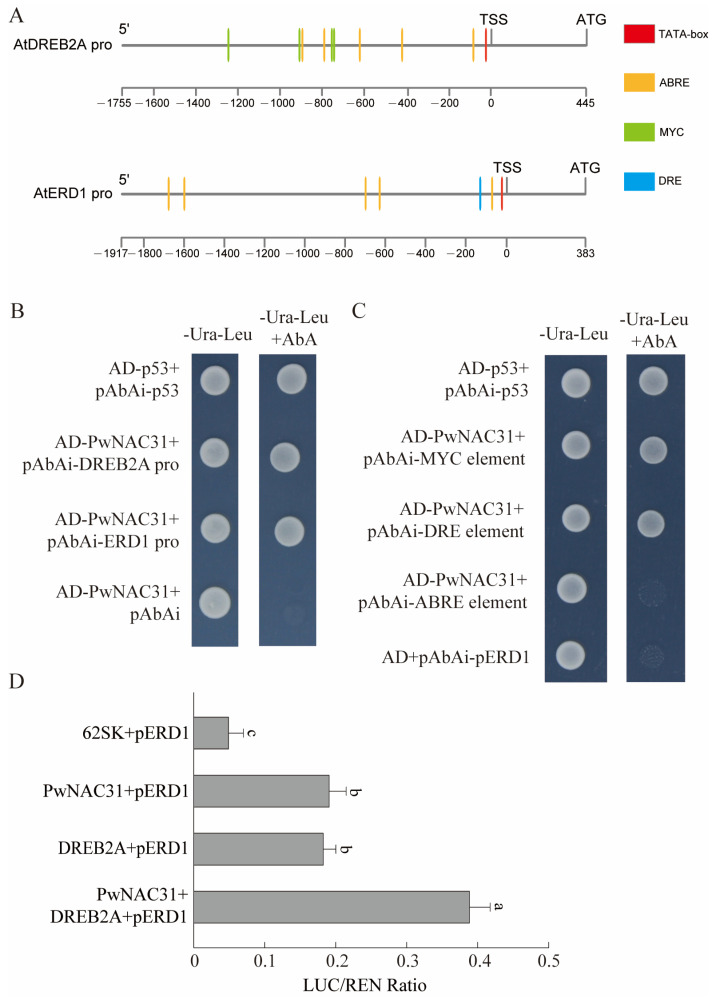
PwNAC31 and DREB2A cooperatively activate *ERD1* transcription. (**A**) The *cis*-acting elements involved in the plant stress response in the promoter region of *AtDREB2A* and *AtERD1*. Five ABRE elements and four MYC elements were detected in the DREB2A promoter, and five ABRE elements and one DRE element were detected in the ERD1 promoter. (**B**) Yeast one-hybrid assays showed the interaction relationship between PwNAC31, *DREB2A* and *ERD1* promoters. Aureobasidin A (AbA) at 500 ng/mL was used to inhibit autoactivation. (**C**) The interaction of PwNAC31 with MYC and DRE element. pGADT7-p53 (“AD-p53”) and pAbAi-p53 were used as positive controls. (**D**) dual-luciferase assay for PwNAC31, DREB2A and ERD1. The error bars indicate ±SD of three independent replications, and the different lowercase letters in Figure (**D**) mean significant differences at 5% level between treatments.

**Figure 8 ijms-25-02037-f008:**
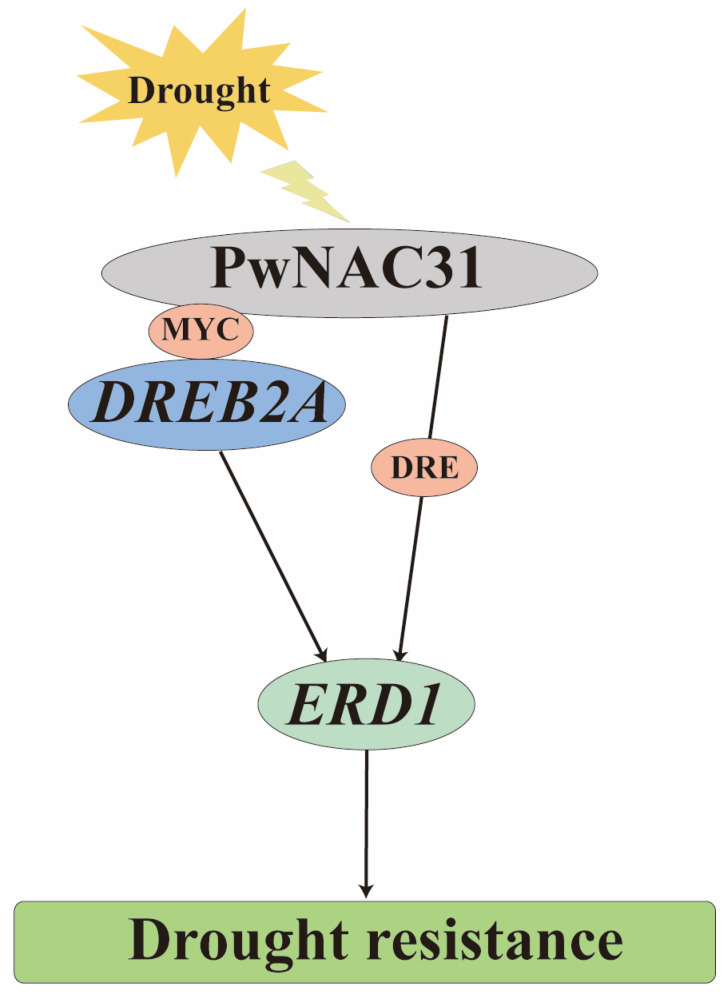
Diagram model of PwNAC31 expression in response to drought stress. PwNAC31 can interact with DREB2A and cooperatively activates the expression of *ERD1*.

## Data Availability

All relevant data can be found within the manuscript and its Appendix A.

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
