# Peer review of "Picea wilsonii NAC31 and DREB2A Cooperatively Activate ERD1 to Modulate Drought Resistance in Transgenic Arabidopsis"

_ijms, 2024, doi:10.3390/ijms25042037_

Round 1
Reviewer 1 Report
Comments and Suggestions for Authors
The authors have done extensive experiments to support their findings. However, I have major comments for the authors:
1. The title of the manuscript shows an ABA independent pathway operates in response to drought resistance in transgenic Arabidopsis. However, the authors have concluded that PwNAC31 positively modulates drought resistance in Picea wilsonii in an ABA-dependent and ABA-independent manner.
2. The authors have mentioned in line 141-143, ‘These results suggested that the expression of PwNAC31 was dramatically induced in response to drought stress, and its regulation mechanism about drought tolerance might be related to ABA signals.
They have also mentioned in line 252-258, ‘Based on the regulatory network of homologous genes of PwNAC31, the expression level of genes responding to drought and ABA pathway-related stress, including ERD1, DREB2A, RD29A, ABI5, ABF3 and NCED3, were examined using RT-qPCR in different Arabidopsis lines after PEG treatment (Figure 6). The results showed that the relative expression of these genes remained low across all lines at normal conditions. The results showed that the expression levels of the drought-responsive genes and ABA-signaling-pathway genes were up-regulated to various extents in the WT and RE lines after PEG treatment compared to the atnac072 mutant.’
Further mentioned in line 280-283, ‘Compared to the negative control, only yeast cells containing combined plasmids of ADPwNAC31 and pAbAi-MYC or -DRE elements could normally grow up on selective medium, demonstrating that PwNAC31 specifically binds to the DRE and MYC elements, but not the ABREs (Figure 7C).’
In line 379-383, the authors mentioned, ‘However, yeast experiments showed that PwNAC31 specifically bound to the DRE and MYC elements, but not the ABREs (Figure 7C). Therefore, we conclude that NAC31 could be regulating drought tolerance mechanism by both ABA-dependent and ABA-independent signaling pathways.
Your results and discussion show that PwNAC31 specifically bound to the DRE and MYC elements, but not the ABREs. So, how can it be regulated by ABA dependent pathway although you have shown induction of ABF3? What is the role of ABI5?
3. What are the other ABA responsive genes that are induced in response to drought stress? Did you look at ABA signaling pathway?
4. In Figure 7A, there should not be 3’ in promoter sequence. It should be TSS.
Comments on the Quality of English LanguageMinor editing is required.
Author Response
Dear reviewer, Thank you very much for your letter and comments about our manuscript (Manuscript ID: ijms-2558674, Title: Picea wilsonii NAC31 and DREB2A cooperatively activate ERD1 to modulate drought resistance in transgenic Arabidopsis). We have checked the manuscript and revised it according to the comments and suggestions. The revised parts of the text were shown in track change. The point-by-point answers to the comments and suggestions were listed as below. |
2. Point-by-point response to Comments and Suggestions for Authors |
Comments 1: The title of the manuscript shows an ABA independent pathway operates in response to drought resistance in transgenic Arabidopsis. However, the authors have concluded that PwNAC31 positively modulates drought resistance in Picea wilsonii in an ABA-dependent and ABA-independent manner. |
Response 1: Thanks for the reviewer’s comments. In the resubmitted manuscript, we have revised the presentation of the results and discussion section of the article. We found that the expression of PwNAC31 was induced by drought and exogenous ABA, and the expression of some drought-responsive genes and ABA pathway-responsive genes was up-regulated after drought treatment. However, it turned out that PwNAC31 could bind to DREB2A and activate the ERD1, but co-expressing PwNAC31 and ABF3 did not activate the expression of ERD1. This figure has been added to Supplemental Figure 3. We will further investigate the regulatory mechanism of PwNAC31 in the ABA signal pathway in future experiments. And we provide a detailed description of this discussion in the revised manuscript, which were shown as below. In addition, PwNAC31 was significantly induced by ABA signals, and our results also confirmed the interaction between PwNAC31 and ABF3. The ABF transcription factors are crucial for ABA signaling and function downstream of various ABA-mediated stress responses [48,49]. Nevertheless, the expression of PwNAC31 promoted strongly the activation of the reporter gene than empty vector in tobacco, while ABF3 limitedly activated LUC expression. What’s more, the simultaneous expression of PwNAC31 and ABF3 did not enhance the activation of the ERD1 promoter more than PwNAC31 alone. These results suggested ABF3 play an extremely limited role in the molecular mechanisms that PwNAC31 responds to drought by activating the expression of ERD1. Combined with the results that there were significant upregulation of the ABA-signaling-pathway genes ABI5 and ABF3 in PwNAC31 transgenic Arabidopsis plants after exposure to drought stress, we speculate that NAC31 may also be regulating the drought tolerance mechanism through the ABA-dependent pathway, with ABF3 or ABI5 involved in this mechanism. (see the lines 375-389 on page 14)
|
Comments 2: The authors have mentioned in line 141-143, ‘These results suggested that the expression of PwNAC31 was dramatically induced in response to drought stress, and its regulation mechanism about drought tolerance might be related to ABA signals. They have also mentioned in line 252-258, ‘Based on the regulatory network of homologous genes of PwNAC31, the expression level of genes responding to drought and ABA pathway-related stress, including ERD1, DREB2A, RD29A, ABI5, ABF3 and NCED3, were examined using RT-qPCR in different Arabidopsis lines after PEG treatment (Figure 6). The results showed that the relative expression of these genes remained low across all lines at normal conditions. The results showed that the expression levels of the drought-responsive genes and ABA-signaling-pathway genes were up-regulated to various extents in the WT and RE lines after PEG treatment compared to the atnac072 mutant.’ Further mentioned in line 280-283, ‘Compared to the negative control, only yeast cells containing combined plasmids of ADPwNAC31 and pAbAi-MYC or -DRE elements could normally grow up on selective medium, demonstrating that PwNAC31 specifically binds to the DRE and MYC elements, but not the ABREs (Figure 7C).’ In line 379-383, the authors mentioned, ‘However, yeast experiments showed that PwNAC31 specifically bound to the DRE and MYC elements, but not the ABREs (Figure 7C). Therefore, we conclude that NAC31 could be regulating drought tolerance mechanism by both ABA-dependent and ABA-independent signaling pathways. Your results and discussion show that PwNAC31 specifically bound to the DRE and MYC elements, but not the ABREs. So, how can it be regulated by ABA dependent pathway although you have shown induction of ABF3? What is the role of ABI5? |
Response 2: Thanks for the reviewer’s comments. We found that the expression of PwNAC31 was induced by drought and exogenous ABA treatment, and PwNAC31 could be regulating drought tolerance by ABA-dependent signaling pathways. Next, predicting the relevant interacting proteins by the STRING website (https://cn. string-db.org/). The results from the previous study showed [1]: the NAC Transcription Factor PwNAC11 Activates ERD1 by Interaction with ABF3 and DREB2A to Enhance Drought Tolerance in Transgenic Arabidopsis. Meanwhile, ERD1 has been reported to be activated by ANAC072, a homologue of PwNAC31, to enhance drought tolerance [2]. Therefore, it is speculated that PwNAC31 positively regulated the expression of DREB2A and its downstream target genes-ERD1, so as to improve the drought resistance of plants. We examined the expression level of genes responding to drought and ABA. The results revealed that the expression of ERD1 and DERB2A changed the most after drought treatment compared to the mutants, while ABF3 only showed an increase at 3h after drought treatment. And the relative expression of ABI5 was lower compared to DREB2A and ERD1. LUC experiments showed that co-expression of ABF3 and PwNAC31 did not activate the expression of ERD1. In contrast, PwNAC31 can combine with the DREB2A and ERD1 promoters, and co-expression of DREB2A and PwNAC31 could activate ERD1 gene expression. In summary, we chose to explore how PwNAC31, DREB2A, and ERD1 work together to improve drought tolerance. Thank you very much for your valuable suggestions. In the future, we will continue to explore the mechanism of the related genes in the ABA signal pathway, including ABF3 and ABI5, on the enhancement of drought tolerance by PwNAC31. Or we will further investigate whether PwNAC31 binds to ABF3 to activate other downstream target genes to enhance drought tolerance. Meanwhile, we enriched the discussion in the resubmitted manuscript. (see the lines 375-388 on page 14)
|
Comments 3: What are the other ABA responsive genes that are induced in response to drought stress? Did you look at ABA signaling pathway? |
Response 3: Thanks for the reviewer’s comments. In this study, we selected ABF3 transcription factors that can interact with PwNAC31. ABF3 transcription factors is crucial for ABA signaling and function downstream of various ABA-mediated stress responses. But what a pity, the simultaneous expression of PwNAC31 and ABF3 did not enhance the activation of the ERD1 promoter more than PwNAC31 alone. We're guessing that ABF3 play an extremely limited role in the molecular mechanisms that PwNAC31 responds to drought by activating the expression of ERD1. Combined with the results that there were significant up-regulation of the ABA-signaling-pathway genes ABI5 and ABF3 in PwNAC31 transgenic Arabidopsis plants after exposure to drought stress. We speculate that PwNAC31 may also be regulating the drought tolerance mechanism through the ABA-dependent pathway, with ABF3 or ABI5 involved in this mechanism. In the future, we will continue to explore the mechanism by which ABA-responsive genes work with PwNAC31 to enhance drought tolerance under drought stress.
|
Comments 4: In Figure 7A, there should not be 3’ in promoter sequence. It should be TSS. |
Response 4: Thanks for the reviewer’s comments. In the revised manuscript, the Figure 7A have been replaced with the correct image. |
- Yu, M.; Liu, J.; Du, B.; Zhang, M.; Wang, A.; Zhang, L., NAC Transcription Factor PwNAC11 Activates ERD1 by Interaction with ABF3 and DREB2A to Enhance Drought Tolerance in Transgenic Arabidopsis. Int. J. Mol. Sci. 2021,22, (13). https://doi.org/10.3390/ijms22136952
- Tran, L.S.; Nakashima, K.; Sakuma, Y.; Simpson, S.D.; Fujita, Y.; Maruyama, K.; Fujita, M.; Seki, M.; Shinozaki, K.; Yamaguchi-Shinozaki, K., Isolation and functional analysis of Arabidopsis stress-inducible NAC transcription factors that bind to a drought-responsive cis-element in the early responsive to dehydration stress 1 promoter. Plant Cell 2004,16, (9), 2481-98. https://doi.org/10.1105/tpc.104.022699

Reviewer 2 Report
Comments and Suggestions for Authors
The research article, titled "Picea wilsonii NAC31 and DREB2A Cooperatively Activate ERD1 to Modulate Drought Resistance in Transgenic Arabidopsis," authored by Yiming Huang et al., investigates the role of the PwNAC31 gene in drought resistance in Picea wilsonii. The study highlights that PwNAC31, when expressed in Arabidopsis, improves drought tolerance. The paper comprises various experiments, such as gene expression analysis, interaction assays, and stress tolerance tests conducted on transgenic Arabidopsis.
Peer Review Comments
Overall Summary:
The paper provides valuable insights into the molecular mechanisms of drought resistance in Picea wilsonii. The authors successfully demonstrate the role of PwNAC31 in enhancing drought tolerance, contributing to the field of plant molecular biology and genetic engineering.
However, several core issues need to be addressed:
Clarity and Depth in Discussion:
The discussion needs to delve deeper into the implications of the findings, particularly about existing literature. A comparison with other NAC transcription factors could better contextualize PwNAC31's uniqueness and efficacy.
Data Interpretation and Analysis:
Clarification is needed in interpreting results, especially in gene expression and functional analysis. Some sentences are unclear, and the meaning could be improved for better understanding. Such as, “our study provides evidence that PwNAC31 positively modulates drought resistance in Picea wilsonii in an ABA-dependent and ABA-independent manner”(line 27-28). “ As shown in Figure. 1A, we found that PwNAC31 and ANAC072(At1g69490) had a higher evolutionary relationship, suggesting that PwNAC31 may have a similar function”(line 119-120).
Data presentation:
It has been noted that there are inconsistencies in the data presentation regarding terms such as "Germination rates" and "Germination percentages" in Figures 3B and C. Additionally, the significance analysis is missing in some cases. Is it possible to clarify these terms, and perhaps provide a visual representation of the data? Furthermore, Figure 4A displays MS and 100 mM mannitol treatment, while Figure 4B shows four treatments.
Broader Range of Drought-Related Traits:
While seed germination rate and root length are good starting points, incorporating additional drought resistance traits, such as leaf water potential, osmotic adjustment, or photosynthetic efficiency, could provide a more comprehensive understanding.
Language and Style:
Minor grammatical and stylistic revisions are needed for clarity and academic language, as some sentences lack the necessary academic tone. For example, “It has been reported that the expression level of NAC genes was induced by drought and ABA treatments [36], so the relative quantitative assay of PwNAC31 was determined under drought and abiotic stress treatments using RT-qPCR. Under drought treatment, PwNAC31 was induced in the late stages (6-12 h) (Figure 1)”(line 133-135). “Meanwhile, to investigate the tissue-specific expression pattern of PwNAC31, the total RNA was extracted from various P. wilsonii, and the results showed that PwNAC31 was expressed in the pollen, root, stem, needle, seed and cone, but had a higher transcript level in fruits than in other tissues(line 138-140).”

Comments on the Quality of English LanguageMinor editing is required.
Author Response
Dear reviewer, Thank you very much for your letter and comments about our manuscript (Manuscript ID: ijms-2558674, Title: Picea wilsonii NAC31 and DREB2A cooperatively activate ERD1 to modulate drought resistance in transgenic Arabidopsis). We have checked the manuscript and revised it according to the comments and suggestions. The revised parts of the text were shown in track change. The point-by-point answers to the comments and suggestions were listed as below. |
2. Point-by-point response to Comments and Suggestions for Authors |
Comments 1: The discussion needs to delve deeper into the implications of the findings, particularly about existing literature. A comparison with other NAC transcription factors could better contextualize PwNAC31's uniqueness and efficacy. |
Response 1: Thanks for the reviewer’s comments. In the revised manuscript, We have added some references on NAC TFs in the second paragraph of the discussion. And we provide a detailed description of this discussion in the revised manuscript, which were shown as below. NAC proteins, as transcription factors, are known to contain a highly conserved NAC domain and play an important role in flower formation and fruit ripening [32], roots growth [33], leaf senescence [34], low-/high-temperature stress [35,36] and drought and flooding stress [37,38,39].(see the lines 311-314 on page 13)
|
Comments 2: Clarification is needed in interpreting results, especially in gene expression and functional analysis. Some sentences are unclear, and the meaning could be improved for better understanding. Such as, “our study provides evidence that PwNAC31 positively modulates drought resistance in Picea wilsonii in an ABA-dependent and ABA-independent manner”(line 27-28). “ As shown in Figure. 1A, we found that PwNAC31 and ANAC072(At1g69490) had a higher evolutionary relationship, suggesting that PwNAC31 may have a similar function”(line 119-120). |
Response 2: Thanks for the reviewer’s comments. In the revised manuscript, “our study provides evidence that PwNAC31 positively modulates drought resistance in Picea wilsonii in an ABA-dependent and ABA-independent manner” has been replaced by “Collectively, our study provides evidence that PwNAC31 activates ERD1 by interaction with DREB2A to enhance drought tolerance in transgenic Arabidopsis.”(see the lines 27-29 on page 1) “As shown in Figure. 1A, we found that PwNAC31 and ANAC072(At1g69490) had a higher evolutionary relationship, suggesting that PwNAC31 may have a similar function” has been replaced by “Phylogenetic analysis showed that PwNAC31 from Picea wilsonii, exhibited a high evolutionary relationship with ANAC072 (At1g69490) from Arabidopsis.” (see the lines 113-114 on page 3)
|
Comments 3: It has been noted that there are inconsistencies in the data presentation regarding terms such as "Germination rates" and "Germination percentages" in Figures 3B and C. Additionally, the significance analysis is missing in some cases. Is it possible to clarify these terms, and perhaps provide a visual representation of the data? Furthermore, Figure 4A displays MS and 100 mM mannitol treatment, while Figure 4B shows four treatments. |
Response 3: Thanks for the reviewer’s comments. I am sorry that this part was not clear in the original manuscript. In our experimental design we wanted to show the dynamic germination process from the first day to the seventh day after sowing through “Germination rates”. The Germination percentages were chosen to be the percentage of germination at the time of seven days of mannitol treatment. Thus, a more complete picture of the effects of simulated drought on different Arabidopsis strains was presented. Then, Supplementary Figure 1 has been modified and replaced with the correct images. Figure 3D has added visual data as support for the results. Thank you for your suggestion, and in order to maintain the consistency of the illustrations, Figure 4A has been replaced with the new images.
|
Comments 4: While seed germination rate and root length are good starting points, incorporating additional drought resistance traits, such as leaf water potential, osmotic adjustment, or photosynthetic efficiency, could provide a more comprehensive understanding. |
Response 4: Thanks for the reviewer’s comments. In the revised manuscript, Supplementary Figure 2 has been added into Supplementary materials. Materials and methods has been added into Section 4.9. We provide a detailed description of this result in the revised manuscript, which were shown as below. After three hours of drought treatment, the leaf water content of the mutant lines was significantly reduced by 40% compared to the WT and RE-2, 4 lines. The drought treatment resulted in a general decrease in chlorophyll content compared to the control, with the mutant having the lowest content (Figure 4E, F). (see the lines 217-221 on page 8) 4.9. Physiology indices measurement Fruit pods length and plant height of Arabidopsis seedlings were measured after four weeks of growth in soil. Several stress damage indices were assessed on seedlings in order to evaluate the effects of drought treatment. Drought-stressed seedlings were weighted at periods of 0, 0.5, 1, 2, and 3 hours at room temperature in order to calculate the relative leaf water content. 80% acetone was used to extract the chlorophyll content which determined with reference to the method of Shah et al. [57] (see the lines 531-536 on page 18)
|
Comments 5: Minor grammatical and stylistic revisions are needed for clarity and academic language, as some sentences lack the necessary academic tone. For example, “It has been reported that the expression level of NAC genes was induced by drought and ABA treatments [36], so the relative quantitative assay of PwNAC31 was determined under drought and abiotic stress treatments using RT-qPCR. Under drought treatment, PwNAC31 was induced in the late stages (6-12 h) (Figure 1)”(line133-135). “Meanwhile, to investigate the tissue-specific expression pattern of PwNAC31, the total RNA was extracted from various P. wilsonii, and the results showed that PwNAC31 was expressed in the pollen, root, stem, needle, seed and cone, but had a higher transcript level in fruits than in other tissues(line 138-140). |
Response 5: Thanks for the reviewer’s comments. English expression has been carefully improved throughout the manuscript. In the revised manuscript, the two sentences and other inappropriate descriptions has been modified and were shown in track change . |

Round 2
Reviewer 1 Report
Comments and Suggestions for Authors
Thank you authors for revising the manuscript.
I do not have any more comments.
Comments on the Quality of English LanguageMinor editing is required.